Mechanical properties of the cuticles of three cockroach species that differ in their wind-evoked escape behavior

Clark Andrew J. 1
Triblehorn Jeffrey D. 1 2 triblehornj@cofc.edu
1 Department of Biology, College of Charleston , Charleston, SC , USA
2 Program in Neuroscience, College of Charleston , Charleston, SC , USA
Oppert Brenda
Electronic publication date: 2014 Jul 31
Publication date: 2014
Volume: 2
Electronic Location ID: e501
Received 2014 Jun 10; Accepted 2014 Jul 13
Copyright: © 2014 Clark and Triblehorn
Copyright year: 2014
Copyright holder: Clark and Triblehorn
License: This is an open access article distributed under the terms of the Creative Commons Attribution License, which permits unrestricted use, distribution, reproduction and adaptation in any medium and for any purpose provided that it is properly attributed. For attribution, the original author(s), title, publication source (PeerJ) and either DOI or URL of the article must be cited.
License URL: https://creativecommons.org/licenses/by/4.0/

Keywords: Blattaria, Insect, Exoskeleton, Cercal system, Predator–prey

Funding: National Center for Research Resources 5 P20 RR016461 National Institute of General Medical Sciences 8 P20 GM103499 Howard Hughes Medical Institute Undergraduate Education Grant College of Charleston, Department of Biology This work was supported by grants from the National Center for Research Resources [grant number 5 P20 RR016461] and the National Institute of General Medical Sciences [grant number 8 P20 GM103499] from the National Institutes of Health as well as a Howard Hughes Medical Institute Undergraduate Education Grant. The College of Charleston Department of Biology also provided research funds to AJC. The funders had no role in study design, data collection and analysis, decision to publish, or preparation of the manuscript.

==============================
The structural and material properties of insect cuticle remain largely unexplored, even though they comprise the majority (approximately 80%) of animals. Insect cuticle serves many functions, including protection against predatory attacks, which is especially beneficial to species failing to employ effective running escape responses. Despite recent advances in our understanding of insect escape behaviors and the biomechanics of insect cuticle, there are limited studies on the protective qualities of cuticle to extreme mechanical stresses and strains imposed by predatory attacks, and how these qualities vary between species employing different escape responses. Blattarians (cockroaches) provide an appropriate model system for such studies. Wind-evoked running escape responses are strong in Periplaneta americana, weak in Blaberus craniifer and absent in Gromphodorhina portentosa, putting the latter two species at greater risk of being struck by a predator. We hypothesized that the exoskeletons in these two larger species could provide more protection from predatory strikes relative to the exoskeleton of P. americana. We quantified the protective qualities of the exoskeletons by measuring the puncture resistance, tensile strength, strain energy storage, and peak strain in fresh samples of thoracic and abdominal cuticles from these three species. We found a continuum in puncture resistance, tensile strength, and strain energy storage between the three species, which were greatest in G. portentosa, moderate in B. craniifer, and smallest in P. americana. Histological measurements of total cuticle thickness followed this same pattern. However, peak strain followed a different trend between species. The comparisons in the material properties drawn between the cuticles of G. portentosa, B. craniifer, and P. americana demonstrate parallels between cuticular biomechanics and predator running escape responses.

Introduction

Despite our long-held fascination with insect cuticle, we have limited understanding about its biomechanical properties (Vincent & Wegst, 2004; Dirks & Taylor, 2012). Insect cuticle is particularly interesting because it is generally strong yet lightweight, and its stiffness (Young’s modulus of elasticity) spans over eight orders of magnitude (Vincent & Wegst, 2004; Klocke & Schmitz, 2011). These characteristics make insect cuticle a biologically inspirational material that could prove beneficial to humans. Recent investigations of the material properties of insect cuticle have used a variety of mechanical testing approaches on various insect body parts to gain a better understanding of how mechanical properties of the cuticle relate to functional morphology (Burrows, 2003; Burrows, Shaw & Sutton, 2008; Dai & Yang, 2010; Dirks & Taylor, 2012; Dirks, Parle & Taylor, 2013), variation in humidity (Schöberl & Jäger, 2006; Klocke & Schmitz, 2011), behavior and general biology (Sun, Tong & Ma, 2008), and composition (Burrows, Shaw & Sutton, 2008; Cribb et al., 2008; Cribb et al., 2010). Insect body parts examined in these studies include the limbs of locusts (Dirks & Taylor, 2012; Dirks, Parle & Taylor, 2013), froghoppers (Burrows, 2003; Burrows, Shaw & Sutton, 2008) and beetles (Sun, Tong & Ma, 2008), the elytra of beetles (Dai & Yang, 2010), the mandibles of grasshoppers (Schöberl & Jäger, 2006), termites (Cribb et al., 2008), and beetles (Cribb et al., 2010), and the sternum in locusts (Klocke & Schmitz, 2011). Though the abdomen comprises a significant part of the body in many insect species, there have been extraordinarily few investigations on the material properties of insect abdominal cuticle (but see Reynolds, 1975; Hackman & Goldberg, 1987).

In addition to being an interface between the insect and its environment, the cuticle serves many functions including: structural support, water loss reduction, respiration facilitation, providing a substrate for multiple sensory receptors, food storage, protection from routine mechanical stresses associated with locomotion and feeding, and protection from parasites and disease (Vincent & Wegst, 2004). Insect cuticle can also function as “body armor”, serving as a morphological active defense mechanism against predatory attacks (Kavaliers & Choleris, 2001). Predatory attacks impose extreme mechanical stresses and strains on insect cuticle. However, there are limited studies on the protective qualities of insect cuticle to such attacks, which may include crushing attacks as well as piercing strikes from teeth and claws.

The cuticle ultimately protects an insect from mechanical damage inflicted by an attacking predator after its passive (i.e., camouflage and cryptic coloration) and other active defenses (i.e., fast predator avoidance behavioral responses) have failed (Kavaliers & Choleris, 2001). Possessing both the behavioral responses effective at eluding predators and cuticle resistant to the most forceful predatory strikes would be most beneficial to the insect. However, morphological protection and behavioral responses may be incompatible with each other (i.e., stronger, and usually thicker, cuticle may impair the ability to perform fast predator avoidance responses) or there may be limited resources during development to produce both stronger cuticle and the anatomical structures adapted for powering rapid escape responses. Therefore, possessing the strongest and thickest varieties of cuticle would be beneficial to insects lacking effective escape responses, while species exhibiting effective escape responses would benefit from the absence of excessively strong and thick cuticle that could impair optimal escape performance. Since insect cuticle has multiple functions, it is difficult to determine how the presence, or absence, of predatory escape responses and the protective function of insect cuticle against predatory attacks have factored into the characteristics of cuticle (i.e., relative contributions of the exocuticule and endocuticle, the microscopic and submicroscopic structure) in different species. However, it is possible to determine the protective biomechanical characteristics of cuticle and compare them across species that differ in their predator avoidance responses.

Cockroaches (Blattaria) provide a good system for such studies since species exhibit diverse predator avoidance responses. Some species, such as the American cockroach Periplaneta americana, exhibit a well-described wind-evoked escape running response mediated by the cercal system (reviewed in Camhi, 1984; Ritzmann, 1984; Comer & Robertson, 2001; Yager, 2010). After detecting wind generated by the attacking predator, P. americana initially responds by rapidly turning away to avoid the initial strike (behavioral latency of 42 ms), followed by continued running. P. americana runs very quickly (up to 1.5 m s−1 or 50 body lengths per second) with high maneuverability (Full & Tu, 1991). Other Blattarian species do not exhibit the response observed in P. americana. When on solid substrates, wind evokes only weak escape running in Blaberus craniifer (Death’s head cockroach). This response is not effective for evading capture (Simpson, Ritzmann & Pollack, 1986) even though this species is capable of moving moderately fast at half the maximum running speed of P. americana (based on studies using Blaberus discoidalis; Full & Tu, 1991). Instead, B. craniifer burrows into softer substrates (when possible) using its head and pronotum, allowing the cockroach to hide either at the approach of a predator or after an attack (Simpson, Ritzmann & Pollack, 1986). Wind does not evoke any escape running in the Madagascan hissing cockroach Gromphadorhina portentosa, which is morphologically more robust than the other two species. However, this species is able to produce a hissing sound generated by expelling air through modified spiracles (Guthrie, 1966; Roth & Hartman, 1967) in response to wind or tactile stimuli. The hissing sound can startle the predator and allow G. portentosa more time to escape. Both B. craniifer and G. portentosa are larger than P. americana (Table 1), which may deter some predators from attacking B. craniifer and G. portentosa. However, the relatively weak or absent wind-mediated escape responses exhibited by B. craniifer and G. portentosa puts these species at greater risk of being struck by a predator relative to P. americana. Therefore, B. craniifer and G. portentosa would benefit from the added protection provided by stronger and thicker cuticle.

Table 1 Summary of cockroach thoracic and abdominal cuticle mechanical properties and morphology.

Measurements of the four biomechanical properties for both thoracic and abdominal cuticle. The thickness measurements of cuticle layers as well as total cuticular thickness are also included for the three cockroach species. Measurements are mean ± s.e.m.

	P. americana	B. craniifer	G. portentosa	
Body length (mm)	
	33.97 ± 3.7	48.46 ± 2.21	57.55 ± 3.61	
Puncture resistance (N)	
Thorax	2.29 ± 0.24	9.67 ± 0.74	30.75 ± 2.21	
Abdomen	1.22 ± 0.09	4.03 ± 0.27	23.49 ± 0.72	
Tensile strength (MPa)	
Thorax	3.64 ± 0.32	11.67 ± 1.80	18.49 ± 2.89	
Abdomen	2.52 ± 0.26	8.05 ± 1.23	17.38 ± 1.09	
Strain energy storage (MJ/m3)	
Thorax	0.103 ± 0.011	0.147 ± 0.036	0.348 ± 0.045	
Abdomen	0.026 ± 0.004	0.114 ± 0.013	0.166 ± 0.037	
Peak strain (%)	
Thorax	8.9 ± 0.8	3.6 ± 0.4	6.3 ± 0.8	
Abdomen	4.2 ± 0.7	7.1 ± 0.6	2.8 ± 0.5	
Total cuticular thickness (µm)	
Thorax	35.83 ± 8.77	75.93 ± 7.66	255.4 ± 14.01	
Abdomen	15.97 ± 1.61	35.38 ± 3.91	126.2 ± 7.62	
Epicuticle thickness (µm)	
Thorax	2.14 ± 0.62	1.48 ± 0.36	4.27 ± 1.35	
Abdomen	1.83 ± 0.09	3.24 ± 0.24	3.79 ± 0.46	
Exocuticle thickness (µm)	
Thorax	19.69 ± 5.42	48.62 ± 5.46	97.11 ± 5.86	
Abdomen	6.9 ± 0.85	18.9 ± 3.2	48.69 ± 3.53	
Endocuticle thickness (µm)	
Thorax	14 ± 3.37	25.83 ± 3.83	154.07 ± 11.49	
Abdomen	7.24 ± 1.23	13.24 ± 1.07	73.72 ± 6.47	

In this study, we investigated the biomechanical properties of the thoracic and abdominal cuticle in these three cockroach species (P. americana, B. craniifer and G. portentosa) that exhibit these different degrees of wind-mediated behavioral responses. While there have been a considerable number of investigations addressing the structure (Jensen et al., 1997; Dennell & Malek, 1956; Kramer & Wigglesworth, 1950) and tanning (Mills, Androuny & Fox, 1968; Koeppe & Mills, 1972; Malex, 1952; Fox, Seed & Mills, 1972; Dennell & Malek, 1955) of cockroach cuticle, the biomechanical properties of Blattarian cuticle have only been measured in the German cockroach Blattella germanica (Czapla, Hopkins & Kramer, 1990). We measured puncture force resistance and three material properties (tensile strength, extensibility, and strain energy storage) to relate the protective qualities of the thoracic and abdominal dorsal cuticle to different types of predatory attacks: (1) tensile strength, extensibility, and strain energy storage provided measures of cuticle resistance to crushing damage; and (2) puncture resistance gauged the amount of protection against piercing damage. Although many factors contribute to the final characteristics of insect cuticle, if protection and predator escape responses were also factors, we predict that species lacking effective wind-evoked escape responses (B. craniifer and G. portentosa) should possess significantly stronger and more puncture resistant cuticle than species that exhibit effective escape responses (P. americana). Furthermore, if the cuticle of B. craniifer and G. portentosa were stronger and exhibited greater tensile strains prior to failure (extensibilities) than that in P. americana, then the cuticle of B. craniifer and G. portentosa should be thicker (assuming similar material composition) and absorb more strain energy per unit volume before fracturing than the cuticle of P. americana.

Materials and Methods

Animals

This study included three adult cockroach (Blattaria) species: Periplaneta americana (Linnaeus, 1758) (Blattidae: Blattinae), Gromphadorhina portentosa (Schaum, 1853) (Blaberidae: Oxyhaloinae), and Blaberus craniifer (Burmeister, 1838) (Blaberidae: Blaberinae) (Fig. 1A). Each species was lab-reared from colonies maintained at the College of Charleston. They were provided cat chow, water, and raised between 24 and 28 °C in 30–60% humidity using a 14:10 day:night cycle.

Figure 1 Cockroach species used in the present study, and the methods used for dissecting, fabricating, testing, and analyzing cuticle samples subjected to tensile and puncture tests.

(A) Photographs of the three cockroach species (left to right): G. portentosa, B. craniifer, and P. americana. All animals were photographed at the same magnification. (B) Photographs (dorsal views) of the three species with their legs and wings removed (no wings in G. portentosa). The thoracic (T2–T3) and abdominal segments (A2–A6) examined are highlighted in light blue. (C and D) Photo of a B. craniifer specimen, with insets showing fabricated samples of abdominal segment cuticles for tensile tests (C) and puncture tests (D). Illustrations to the right demonstrate how cuticle samples were positioned in the apparatuses for tensile testing (C) and puncture testing (D). (E and F) Representative data sets collected from a B. craniifer abdominal segment sample, with methods for measuring peak tensile stress (tensile strength), peak strain (extensibility), and strain energy (E), and puncture force (F). Scale bars = 1.0 cm.

Cuticle sample preparation

Prior to sample preparation, cockroaches were placed in a freezer (−30 °C) for 5–10 min in order to reduce activity. The animal was then pinned to a Sylgard dish (WPI, Inc) after removing the legs (P. americana, G. portentosa, and B. craniifer) and wings (P. americana and B. craniifer) (Fig. 1B). The dorsal cuticle from the mesothoracic (T2) and metathoracic segments (T3), as well as abdominal segments two through six (A2–A6), were removed. Samples were taken from these segments and prepared for mechanical testing by cutting the cuticle into either dumbbell shapes for tensile testing or squares for puncture testing (Fig. 1C).

Mechanical testing

Fabricated cuticle samples from all cockroach species were subjected to quasi-static uniaxial puncture tests and tensile tests to failure, and the testing of all cuticles occurred immediately after excision from the animals. An Imada EMX-275 (Northbrook, IL, U.S.A.) motorized vertical testing stand, equipped with an Imada ZP-11 (50 N capacity) force gauge (Northbrook, IL, U.S.A.) and a Mitutoyo Digimatic 570-244 height gauge (Aurora, IL, U.S.A.), was used for conducting tensile and puncture tests on the abdominal (A2–A6) and thoracic (T2 and T3) cuticle segments of 40 P. americana (32 for tensile tests; eight for puncture tests), 22 G. portentosa (16 for tensile tests; six for puncture tests), and 19 specimens B. craniifer (13 for tensile tests; six for puncture tests).

Prior to tensile tests, dumbbell-shaped cuticle samples were clamped within the standard serrated grips provided with the test stand. The bottom grip was stationary, and the top grip was connected to the force gauge affixed to the test stand’s vertical linear actuator (Fig. 1C). Tensile data sets in our analysis were only collected from cuticle samples that exhibited mechanical failure in the narrowest portion of the dumbbell. For the puncture tests, a two mm diameter pin was attached to the force gauge and driven into the square-shaped cuticle samples, which were positioned between two five mm thick square acrylic plates (three cm × three cm). Four mm holes were drilled into the centers of both acrylic plates, which allowed passage of the two mm puncture pin through the cuticle sample (Fig. 1C). The bottom acrylic plate was glued to a hollow metal cylinder, which served as the base of the puncture apparatus and also allowed passage of the puncture pin if it passed below the bottom acrylic plate during testing. Both top and bottom acrylic plates were clamped together on their right and left ends with binder clips to prevent slippage of cuticle samples during puncture testing.

All samples were strained at a rate of 1.5 mm/min, and force and extension data were recorded at 2.0 Hz. Peak puncture forces (FP) were determined from raw force–extension curves from samples subjected to puncture tests, in which the FP equaled the maximum applied force prior to failure (Fig. 1D). Prior to analyzing data sets collected from tensile tests, stress (σ) and strain (ε) were derived from force (F) and extension (L) measurements. Stress was calculated as: (1) σ=F/CSA,

where the cross-sectional area (CSA) of each sample, which was the area orthogonal to the applied tensile force (F), was calculated as the product of sample width and thickness. Strain was calculated as: (2) ε=ΔL/L0,

where ΔL represented the change in length during the tensile test, and the initial sample length prior to testing (L0) equaled the grip separation. All cuticle samples exhibited J-shaped stress–strain curves, which included a shallow-sloped curvy toe region at lower strains followed by a steeper-sloped linear region at greater strains (Fig. 1D). Peak stress (strength) was defined as the maximum applied stress prior to failure, and peak strain (extensibility) was defined as the strain at failure (Fig. 1D). Strain energy storage (work of extension) was measured as the area under the stress–strain curve using Prism 5 or 6 (GraphPad Software, Inc, San Diego, CA) (Fig. 1D).

Cuticle morphology

Histological sectioning, staining, and thickness measurements were performed on the metathoracic (T3) and fourth abdominal (A4) segments from P. americana, G. portentosa, and B. craniifer (five specimens per species). The cuticles were fixed in 10% buffered formalin overnight. Cuticles were dehydrated with ethanol, embedded in paraffin and sectioned at 5 µm. In all animals, abdominal and thoracic segments were sectioned in the transverse plane of the animal at approximately 50% of the segment length between the anterior and posterior edges. Sections were stained with hematoxylin and eosin, which facilitated the identification of the epicuticle and the exocuticle and endocuticle layers of the procuticle.

Total cuticule thickness measurements as well as the thickness of the epicuticle, exocuticle, and endocuticle layers comprising the cuticle were performed on digital images using ImageJ version 1.48 (W.S. Rasband, U.S. National Institutes of Health, Bethesda, MD, USA, http://imagej.nih.gov/ij/, 1997–2014). Digital images were acquired using a Qcolor3 Olympus digital camera on an Olympus Bx50 microscope using QCapture Pro 6.0.

Statistical analysis

All statistics were performed using Prism 5 or 6 (GraphPad Software, San Diego, CA). Parametric statistical tests (either unpaired one-way ANOVA or t-test) were performed on data with equal variances as determined by performing Bartlett’s test for equal variances. Otherwise, nonparametric tests (either Mann–Whitney or Kruskal–Wallis) were used. Post-hoc tests involved Tukey’s multiple comparison tests for parametric data and Dunn’s multiple comparison tests for nonparametric data.

Results

Mechanical testing of cockroach thoracic and abdominal cuticle

We tested four mechanical properties of thoracic and abdominal cockroach cuticle relevant for protection against two different types of attack: tensile strength, peak strain, and strain energy storage, as measures of resistance to crushing; and puncture force resistance, as a measure of resistance to piercing (e.g., biting). Compressive loading produced by a successful crushing attack increases hydrostatic pressure within the prey’s body cavity, which induces tensile stresses and strains that are resisted by the prey’s cuticle until failure. During stretching, the cuticle absorbs the strain energy induced by these tensile stresses and strains until failure. In contrast to squashing or crushing, successful piercing attacks delivered by dentition or claws impose puncture loads that breach the prey’s cuticle at the sites of contact. Within each species, tensile strength, strain energy storage, peak strain, and puncture force resistance did not differ significantly between mesothoracic (T2) and metathoracic (T3) segments or between abdominal segments (A2–A6). Therefore, we pooled T2 and T3 for the thoracic data and A2–A6 for the abdominal data.

G. portentosa thoracic and abdominal cuticle had the greatest resistance to puncture force (thoracic: 30.75 ± 2.21 N; abdomen: 23.49 ± 0.72 N; mean ± s.e.m.) while P. americana had the weakest (thoracic: 2.29 ± 0.24 N; abdomen: 1.21 ± 0.09 N) (Fig. 2A and Table 1). B. craniifer cuticle had peak puncture forces closer to P. americana (thoracic: 9.67 ± 0.74 N; abdominal: 4.03 ± 0.27 N). Puncture force resistance differed significantly across species for the abdominal (Kruskal–Wallis test: H2 = 84.59, p < 0.0001) and thoracic cuticle (Kruskal–Wallis test: H2 = 33.65, p < 0.0001). In all species, the thoracic cuticle was more puncture resistant than the abdominal cuticle (Mann–Whitney U-tests, P. americana: U(38,15) = 96, p = 0.0002; B. craniifer: U(30,12) = 13, p < 0.0001; G. portentosa: U(30,12) = 70, p = 0.0023).

Figure 2 Comparative data sets from puncture and tensile tests on abdominal (A, C, E, H) and thoracic cuticle segments (B, D, F, G) from P. americana, G. portentosa, and B. craniifer.

Mean puncture forces recorded from the abdominal cuticle segments (pooled data from A2 to A6) (A) and thoracic segments (pooled data from T2 and T3) (B). Mean peak tensile stress (tensile strength) of the abdominal segments (pooled data from A2 to A6) (C) and thoracic segments (pooled data from T2 and T3) (D). Mean strain energy storage of the abdominal segments (pooled data from A2 to A6) (E) and thoracic segments (pooled data from T2 and T3) (F). Mean peak strain of the abdominal segments (pooled data from A2 to A6) (G) and thoracic segments (pooled data from T2 and T3) (H). All data sets shown in the right columns are plotted to the same scale and units as those shown in their respective left columns. Error bars in all graphs are standard error of the mean (s.e.m.). ∗, <0.05; ∗∗, <0.01; ∗∗∗, <0.001; n.s., not significant.

G. portentosa thoracic and abdominal cuticle also had the greatest tensile strength (thoracic: 18.49 ± 2.89 MPa; abdominal: 17.38 ± 1.09 MPa) while P. americana cuticle had the weakest (thoracic: 3.64 ± 0.32 MPa; abdominal: 2.52 ± 0.26 MPa) (Fig. 2B and Table 1). Tensile strength for B. craniifer cuticle fell close to directly between these species (thoracic: 11.67 ± 1.80 MPa; abdominal: 8.05 ± 1.23 MPa). Tensile strength was significantly different between the abdominal cuticle for all species examined (Kruskal–Wallis test: H2 = 64.7, p < 0.0001) (Fig. 2B). The thoracic tensile strength across some species was significantly different (Kruskal–Wallis test: H2 = 34.13, p < 0.0001). The thoracic cuticle of P. americana was significantly weaker than that of B. craniifer and G. portentosa. Although not statistically different between B. craniifer and G. portentosa, thoracic cuticle tensile strengths followed the same pattern across species as for the abdominal cuticle (Fig. 2B). Statistically, the thoracic cuticle was significantly stronger than the abdominal cuticle of P. americana (unpaired t-test: t66 = 2.769, p = 0.0073), but abdominal and thoracic cuticle had similar tensile strengths for B. craniifer (unpaired t-test: t52 = 1.627, p = 0.1099, ns) and G. portentosa (Mann–Whitney U-test: U37,18 = 319.0, p = 0.8087, ns).

G. portentosa thoracic and abdominal cuticle stored the greatest amount of strain energy per volume (thoracic: 0.348 ± 0.045 MJ/m3; abdominal: 0.166 ± 0.037 MJ/m3) while P. americana stored the least (thoracic: 0.103 ± 0.011 MJ/m3; abdominal: 0.026 ± 0.004 MJ/m3) (Fig. 2C and Table 1). Strain energy storage of B. craniifer cuticle fell between these species (thoracic: 0.147 ± 0.036 MJ/m3; abdominal: 0.114 ± 0.013 MJ/m3). Strain energy storage for the abdominal cuticle of P. americana was significantly smaller than for the abdominal cuticle of B. craniifer and G. portentosa, but not significantly different between the abdominal cuticle of B. craniifer and G. portentosa (Kruskal–Wallis test: H2 = 58.85, p < 0.0001) (Fig. 2C). The thoracic cuticle of G. portentosa stored significantly more energy than the thoracic cuticle of P. americana and B. craniifer, but not significantly different between the thoracic cuticle of P. americana and B. craniifer (Kruskal–Wallis test: H2 = 21.47, p < 0.0001) (Fig. 2C). Though strain energy storage was not statistically different between the thoracic cuticle of B. craniifer and P. americana or between the abdominal cuticle of B. craniifer and G. portentosa, the data sets for strain energy storage followed the same pattern across species as for the tensile strength and puncture force resistance (Figs. 2A–2C and Table 1).

The pattern of interspecies differences for peak strain differed from the pattern observed for tensile strength, puncture force, and strain energy (Fig. 2D and Table 1). Peak strain of B. craniifer abdominal cuticle was 7.1% of L0, which was significantly larger than the strains recorded for the abdominal cuticle of P. americana (4.2% of L0) and G. portentosa (2.8% of L0) (one-way ANOVA: F2,109 = 14.18, p < 0.0001) (Fig. 2D). Conversely, peak strain of B. craniifer thoracic cuticle (3.6% of L0) was significantly smaller than the strains measured for the thoracic cuticle of P. americana (8.9% of L0) and G. portentosa (6.3% of L0) (Kruskal–Wallis test: H2 = 25.00, p < 0.0001) (Fig. 2D). Peak strains for the abdominal and thoracic cuticles of P. americana did not significantly differ from those of G. portentosa (Fig. 2D).

Cuticle morphology

We performed histology on the thoracic and abdominal cuticle of the three cockroach species to measure total cuticular thickness as well as the relative thicknesses of the epicuticle, exocuticle, and endocuticle layers that comprise the exoskeleton. Hematoxylin and eosin staining of the histological sections of cuticle samples were useful for distinguishing and measuring the exocuticle and endocuticle layers of the procuticle. Transverse sections of the thoracic and abdominal cuticle samples in P. americana, B. craniifer, and G. portentosa demonstrated similar tissue arrangement in both layers of the procuticle (Fig. 3). In the endocuticle, parallel tissue layers were oriented horizontally in the endocuticle while the tissue layers in the exocuticle were oriented semi-vertically.

Figure 3 Comparative morphology of the cuticle in the cuticle of G. portentosa, B. craniifer, and P. americana, and methods used for measuring thickness of the cuticle and its components.

(A) is a photograph of a transverse section of a hematoxylin and eosin stained A4 abdominal segment from a G. portentosa specimen at 40X magnification with different cuticular layers indicated. (B) Methods for sectioning and measuring cockroach cuticles. In all animals, abdominal and thoracic segments were sectioned in the transverse plane of the animal at approximately 50% of the segment length between the anterior and posterior edges, as indicated by the blue dashed line. ENDO, endocuticle; EPIC, epicuticle; EPID, epidermis; and EXO, exocuticle. (C–E) H&E stained histological sections of the A4 abdominal cuticle from specimens of G. portentosa (C), B. craniifer (D), and P. americana (E). For each specimen, C–E are photos taken at 10X magnification and the insets (A, B) are photos taken at 40X magnification. Scale bar = 25 µm for top (40X) photos; 100 µm for bottom (10X) photos.

G. portentosa possessed the thickest thoracic and abdominal cuticle (thoracic: 255.4 ± 14.01 µm; abdominal: 126.2 ± 7.62 µm; mean ± s.e.m.), followed by B. craniifer (thoracic: 75.93 ± 7.66 µm; abdominal: 35.38 ± 3.91 µm) with P. americana possessing the thinnest cuticle of the three species (thoracic: 35.83 ± 8.77 µm; abdominal: 15.97 ± 1.61 µm) (Figs. 4A and 4B; Table 1). The contribution of these layers to the total cuticular thickness, particularly the exocuticle and endocuticle, varied both between abdominal and thoracic sections within and between species (Figs. 4C and 4D). The exocuticle of B. craniifer comprised the largest percentage of both the abdominal (52%) and thoracic cuticle (64%). In P. americana, the exocuticle also comprised the largest percentage of the thoracic cuticle (54%), but the exocuticle and endocuticle contributed equally to the abdominal cuticle (43–45%). The exocuticle comprised a larger percentage of the thoracic cuticle than the abdominal cuticle in both B. craniifer and P. americana. In G. portentosa, the endocuticle layer comprised the largest percentage of the total thickness in both the abdominal (58%) and thoracic cuticle (60%) (Figs. 4C and 4D; Table 1). The relative contribution of each layer to the total cuticular thickness was similar for both abdominal and thoracic cuticle in G. portentosa.

Figure 4 Comparative morphology of the abdominal and thoracic cuticle segments from P. americana, G. portentosa, and B. craniifer.

(A) Mean thickness of the abdominal cuticle segments (pooled data from A4). (B) Mean thickness of the thoracic cuticle segments (pooled data from T2). (C) Relative thickness of the epicuticle, exocuticle, and endocuticle, expressed as percentages of total cuticular thickness from the abdomens of all three cockroach species. (D) Relative thickness of the epicuticle, exocuticle, and endocuticle, expressed as percentages of total cuticular thickness from the thoraces of all three cockroach species. Data come from five individuals per species. Error bars in all graphs are standard error of the mean (s.e.m.).

Discussion

This study examined four biomechanical properties of insect abdominal and thoracic cuticle in three cockroach species (P. americana, B. craniifer, and G. portentosa) that exhibit different degrees of wind-mediated behavioral escape responses. Although many factors contribute to the characteristics of insect cuticle, protection and predator escape responses could be two of these factors. Considering this possibility, we made predictions that the two species that do not exhibit effective wind-evoked escape running responses (B. craniifer and G. portentosa) would possess cuticle that is thicker and more resistant to puncture as well as tensile stress and strain than P. americana, which does exhibit an effective running response. The bases for these predictions were: (1) B. craniifer and G. portentosa would be more susceptible to direct attacks from predators (i.e., biting, piercing, crushing) and benefit from more protection in the form of stronger cuticle and (2) P. americana would possess weaker cuticle since their escape response would protect this species from predator strikes and because stronger cuticle could hamper maneuverability and the effectiveness of escape responses. Our results demonstrated a continuum across the three cockroach species in tensile strength, strain energy storage, puncture force resistance, and total cuticular thickness. As we predicted, P. americana had the weakest and thinnest cuticle while G. portentosa had the strongest and thickest. Thoracic and abdominal cuticles in G. portentosa were more puncture resistant and stored more energy per volume than P. americana. The tensile strength, strain energy storage, and puncture resistance in thoracic and abdominal cuticles of B. craniifer fell between those in G. portentosa and P. americana.

Mechanics and morphology of Blattarian cuticle: patterns with protection and escape behaviors

Tensile strength, peak strain (also known as “extensibility”), and strain energy storage (also known as “work of extension” or “toughness”) are material properties that can describe the physical limits of the cuticle before mechanical failure (i.e., cracking or fracturing). In the case of cockroach cuticle, these properties can provide a window to the mechanical situations associated with successful and unsuccessful crushing attacks delivered by potential predators. Tensile strength is the maximum stress cuticle can experience before breaking. Peak strain is the maximum strain the cuticle can experience before breaking. Strain energy storage is the potential mechanical energy absorbed per unit of cuticle when work is performed to extend and break it. We also measured puncture force resistance, which is not commonly recorded in studies on insect cuticle, but is a particularly important biomechanical property because it pertains to piercing assaults related to biting and clawing attacks from predators. This study is one of very few to demonstrate puncture resistance in insect cuticle (but see Roseland, Kramer & Hopkins, 1987; Czapla, Hopkins & Kramer, 1990). Puncture resistance values in a given species may reflect morphological (e.g., shape and size of teeth, fangs, and claws) and functional (e.g., biting or clawing forces) characteristics of potential predators to which the cuticle is adapted to defend against.

For three of the biomechanical properties (tensile strength, strain energy storage, and puncture force resistance), a consistent pattern emerged of a continuum across the three species where G. portentosa cuticle had the strongest cuticle and P. americana the weakest, with B. craniifer somewhere between the two (Figs. 2A–2C). This result was consistent with our predictions that the two species where wind does not elicit effective escape running responses (G. portentosa and B. craniifer) would be more susceptible to successful strikes by predators and would benefit from the protection conveyed by stronger cuticle. G. portentosa is the most susceptible of the three species as wind may elicit hissing responses to startle predators but no running escape response. Wind does not elicit escape running in B. craniifer that allows the species to escape from predators. However, wind does elicit burrowing responses that may be more effective than the defense response of G. portentosa but not as effective as escape running in P. americana. Thus, the continuum of protection provided by the material properties of the cuticle in these three species is consistent with the continuum of protection provided by the behavioral responses.

Abdominal and thoracic cuticle thickness measured from histological samples showed a continuum across species (Fig. 4), similar to that observed for tensile strength, strain energy storage, and puncture resistance. However, as stress–strain data are normalized to sample dimensions, this similarity suggests that the species differences for these three material properties might be due to cuticular composition instead of total thickness. This warrants further work to assess possible relationships between the cuticular composition and material properties in P. americana, B. craniifer, and G. portentosa. Cuticle thickness was not scaled to body size since body sizes differed by less than two times between any species. However, G. portentosa thoracic and abdominal cuticle was over three times thicker than B. craniifer cuticle and over seven times thicker than P. americana while B. craniifer cuticle was over twice the thickness of P. americana.

The organization of the abdominal and thoracic cuticle was similar between P. americana and B. craniifer, with the cuticle consisting of a larger or equal percentage of exocuticle relative to endocuticle (Figs. 4C and 4D). However, the abdominal and thoracic cuticle of G. portentosa was very different from the other two species in that the endocuticle comprised a larger percentage of the cuticle compared to the exocuticle (Figs. 4C and 4D). Neither the organization of cuticle nor total cuticular thickness can account for the different trend observed in the peak strain results (Fig. 2D). However, one possible explanation for the higher extensible abdominal cuticle in B. craniifer (7% of L0), relative to that in P. americana (4% of L0) and G. portentosa (3% of L0), is that this could be an adaptation for effective burrowing behaviors and maneuvering in tight spaces.

Many biomaterials, like the cuticles of cockroaches and other insects, are composite materials comprised of multiple material constituents with different physical and chemical properties (Vincent & Wegst, 2004). In addition to an epicuticle and epidermis, insect cuticle possesses a procuticle layer that accounts for a significant percentage of the total cuticular thickness, and is the cuticular layer most important for negotiating mechanical stresses and strains. The procuticle includes exocuticle layers and endocuticle layers, by which the exocuticle is the sclerotized layer that grants the cuticle its rigid characteristics while the endocuticle is the unsclerotized layer that grants the cuticle its flexible characteristics (Wigglesworth, 1948; Dennell & Malek, 1956; Filshie, 1982). Although G. portentosa cuticle contains a larger percentage of endocuticle compared to the other two species (Figs. 4C and 4D), the exocuticle in G. portentosa cuticle is actually thicker than the entire cuticle of P. americana or B. craniifer (Table 1), which could account for the higher puncture resistance values for G. portentosa.

For P. americana, the initial turn component of the escape running response allows the animal to avoid the initial strike from a predator while the continued running removes the animal from further attacks. Speed is important for both components of the response and P. americana is one of the fastest invertebrates tested (Bell, Roth & Nalepa, 2007), able to cover 50 body lengths per second at top speed, aided by incorporating an aerial phase when running (Full & Tu, 1991). Although not as fast as P. americana, the maximum running speed of B. discoidalis, closely related to B. craniifer used in the current study, is half the maximum speed of P. americana and runs more awkwardly (Full & Tu, 1991). Conversely, G. portentosa is not a runner and has been described as a cockroach adapted for climbing behaviors instead of running (Bell, Roth & Nalepa, 2007). It is not clear whether thicker cuticle that provides more protection would hamper running ability; could P. americana possess stronger cuticle and be able to run just as fast and escape just as effectively? This is an interesting question for future studies, but the thinner cuticle in P. americana could be related to one or several other functions the exoskeleton serves besides protection and structural support for movement. Regardless of the primary reason for P. americana’s thinner, less protective cuticle, the ability to escape from predators with great speed is most necessary for this species compared to B. craniifer and G. portentosa.

Cockroach predators and protection provided by the cuticle

Blattarians are potential prey for both invertebrate (e.g., spiders, mantids, centipedes, and parasitic wasps) and vertebrate insectivorous animals (e.g., fish, amphibians, reptiles, birds and mammals). The puncture resistance properties of Blattarian cuticle would protect against cuticle penetration by spider fangs, centipede claw attacks and parasitic wasp stingers as well as pecking attacks from birds (i.e., chickens, quails, pigeons). The other cuticular properties (tensile strength, extensibility, and strain energy storage) maintain cuticle integrity during crushing attacks from both invertebrates (i.e., the forelegs of praying mantids) and vertebrates (i.e., the mouthparts of certain fish, lizards and mammals).

If a predator initially fails to subdue its prey (i.e., by injecting venom), it will make additional attempts to consume its prey, during which time the insect’s cuticle can still provide protection. Active prey are difficult to secure and consume and several examples show that predatory animals often mishandle active prey, which provides additional opportunities for the prey to escape. Bats capturing praying mantids on the wing are able to consume the mantis with relative ease if the mantis is eaten head and forelegs first. However, if transferred to the mouth abdomen first, the bat often drops its prey since the mantis is able to use its raptorial forelegs to strike at the bat’s face (Triblehorn, 2003). In another example, dragonfly nymphs capture prey by rapidly extending their labium to impale the prey, which is then retracted to enable the dragonfly to consume the prey. Herberholz, Sen & Edwards (2004) found that captured juvenile crayfish can produce “tail-flips” (rapid flexions of their abdomen) to free themselves from the labium before the dragonfly is able to consume them. Cuticle with greater resistance to crushing and puncture attacks can provide more opportunities for the predator to mishandle the prey and minimize damage in the event the prey is able to escape in this manner.

Comparative mechanics of Blattarian cuticle and other materials

Tensile strengths observed in the thoracic and abdominal cuticles of P. americana (∼3 MPa), B. craniifer (∼10 MPa), and G. portentosa (∼20 MPa), occur within the range of tensile and compressive strengths documented in different insect species (e.g., Ashby et al., 1995; Chen et al., 2002), and various natural elastomers (e.g., leathers, cartilage, and skin) as well as natural polymer composites (e.g., ligaments, tendon, and wool) (Ashby et al., 1995). Cuticular tensile strength and strain energy storage per volume (work of extension, or toughness) are especially important material properties to measure when assessing extreme mechanical stresses, strains, and failure resulting from successful predatory strikes. Though not as commonly recorded as stiffness, cuticular tensile strength has been measured in a variety of insect cuticle, including tibiae, elytra, and hindwings (Table 2). Values for strain energy storage recorded in the cuticles from P. americana (0.03–0.10 MJ/m3), B. craniifer (0.11–0.15 MJ/m3), and G. portentosa (0.17–0.35 MJ/m3), are comparable to those in several biological and synthetic materials (Table 2). Strain energy storage per volume in these Blattarian cuticles is one order of magnitude greater than that in ancient iron but more similar to records from arterial wall, mussel shell, dry yew wood, and bronze, and one to two orders of magnitude less than records from modern spring steel, tendon, and rubber.

Table 2 Mechanical properties of insect cuticle and other non-cuticular materials from previous experiments.

Summary of results from previous experiments testing the mechanical properties of cuticle from other insect species as well as several other biological and synthetic materials.

Species	Common name	Body part	Biomechanical property	Measurement	Reference	
Allomyrina
dichotoma	Japanese
rhinoceros
beetle	Elytra	Tensile strength	130 MPa	Sun & Bhushan (2012)	
Hindwings	Tensile strength	5–50 MPa	Jin et al. (2009)	
Copris ochus	Horned
dung beetle	Elytra	Tensile strength	1200 MPa	Sun et al. (2010)	
Pachynoda
sinuata	Rose chafer
beetle	Elytra	Tensile strength	40–110 MPa	Hepburn & Ball (1973)	
Tenebrio molitor	Darkling
beetle	Elytra	Tensile strength	6–30 MPa	Lomakin et al. (2011)	
Tribolium
castaneum	Red rust
flour beetle	Elytra	Tensile strength	4–10 MPa	Lomakin et al. (2011)	
	Peak puncture force	0.01–0.05 N	Roseland, Kramer & Hopkins (1987)	
Schistocerca	Locust	Tibiae	Tensile strength	95 MPa	Jensen & Weis-Fogh (1962)	
Rhodnius	Kissing bug	Abdominal
cuticle	Peak strain	25–30%	Reynolds (1975)	
Blattella germanica	German
cockroach	Pronotum	Peak puncture force	0.2–1 N	Czapla, Hopkins & Kramer (1990)	
Non-cuticular
materials						
Ancient iron			Strain energy
storage per volume	0.01 MJ/m3	Gordon (1978)	
Arterial wall			Strain energy
storage per volume	0.5 MJ/m3	Gordon (1978)	
Mussel shell			Strain energy
storage per volume	0.5 MJ/m3	Gordon (1978)	
Dry yew wood			Strain energy
storage per volume	0.5 MJ/m3	Gordon (1978)	
Bronze			Strain energy
storage per volume	0.6 MJ/m3	Gordon (1978)	
Modern spring steel			Strain energy
storage per volume	1.0 MJ/m3	Gordon (1978)	
Tendon			Strain energy
storage per volume	2.8 MJ/m3	Gordon (1978)	
Rubber			Strain energy
storage per volume	10.0 MJ/m3	Gordon (1978)	

The J-shaped stress–strain curves observed in these Blattarian abdominal and thoracic cuticles resembled those observed in the abdominal cuticle in Rhodnius larvae (the kissing bug, Hemiptera) (see Reynolds, 1975). However, larval Rhodnius abdominal cuticle can stretch to 25–30% of their original length (Reynolds, 1975), with peak strains that are over one order of magnitude greater than the peak strains we measured in the abdominal cuticles of P. americana (4% of L0), B. craniifer (7% of L0), and G. portentosa (3% of L0). Instead, cuticular peak strains in these Blatarrian species are more similar to peak strains recorded from other insect cuticle, which range from one to two percent of L0 (Vogel, 2003). Moreover, the peak strains recorded in P. americana, B. craniifer, and G. portentosa could be similar to rigid cuticle found in locust (Orthoptera) tibiae (e.g., Dirks & Taylor, 2012) and beetle (Coleoptera) elytra (e.g., Lomakin et al., 2011), or fall somewhere in between these rigid cuticles and soft cuticle (e.g., Reynolds, 1975), but this cannot be determined since those particular studies on rigid cuticle did not include peak strain values.

Cuticular puncture resistance was 25–30 N in G. portentosa, approximately 4–10 N in B. craniifer, and 1–3 N in P. americana (Fig. 2A). The peak puncture forces observed in P. americana, which was the least puncture resistant of three cockroach species, were one to two orders of magnitude greater than the maximum puncture resistances recorded from the elytra of T. castaneum (maximum = 0.05 N, Roseland, Kramer & Hopkins, 1987) and the thoracic cuticle of Blattella germanica (German cockroach) (maximum = 0.8 N; Czapla, Hopkins & Kramer, 1990).

Testing fresh cuticle samples

The current experiment used fresh cuticle samples, which is noteworthy since there are very limited data sets on the mechanics of fresh cuticle (but see Dirks & Taylor, 2012). Most studies involved approaches that require preliminary processing, such as drying and coating of cuticle samples, which alters the biomechanical properties of insect cuticle (Schöberl & Jäger, 2006; Klocke & Schmitz, 2011; Dirks & Taylor, 2012). Prolonged time periods between excision and testing desiccates the cuticle, thus inducing additional cross-linking between reinforcing chitin and proteins comprising the cuticle, which artificially enhances cuticle strength (Vincent, 2009; Klocke & Schmitz, 2011; Dirks & Taylor, 2012). Drying also affects the properties of other biomaterials, including bone (Nyman et al., 2006), squid beaks (Miserez et al., 2008), and equine hooves (Bertram & Gosline, 1987). Therefore, the minimal time between removal of the cuticle from the animals and mechanical testing reduces desiccation of samples and increases the likelihood that our data sets are similar to in situ properties (Dirks & Taylor, 2012).

Supplemental Information

Supplemental Information 1 Cuticle histology dataset

Photos of cuticles and measurements.

Click here for additional data file.

Supplemental Information 2 Raw data from biomechanical cuticle tests

Raw data from each of the tensile and puncture force tests for the three cockroach species used in this manuscript. The tensile tests were used to obtain tensile strength, strain energy storage, and peak strain results and puncture force tests were used to calculate puncture force.

Click here for additional data file.

The authors would like to thank Margaret Romano and the Medical University of South Carolina (MUSC) Histology Core Facility for the preparation, sectioning and staining of the cuticle samples used for the morphological measurements.

Additional Information and Declarations

Competing Interests

Author Contributions

The authors declare there are no competing interests.

Andrew J. Clark and Jeffrey D. Triblehorn conceived and designed the experiments, performed the experiments, analyzed the data, contributed reagents/materials/analysis tools, wrote the paper, prepared figures and/or tables, reviewed drafts of the paper.

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
