# Peer review of "Mechanical properties of the cuticles of three cockroach species that differ in their wind-evoked escape behavior"

_PeerJ, doi:10.7717/peerj.501_

## Round 0.1 · original submission · Minor Revisions

· Academic Editor

Minor Revisions

The reviewers feel that the paper is a valuable contribution and scientifically sound, but they have raised a few questions that, when addressed, will greatly improve the manuscript. I ask that you consider these suggestions and address the comments in a revision.

Reviewer 1 ·

Basic reporting

No Comments

Experimental design

Table 1: Some editing problems (white boxes instead of +/-)

Validity of the findings

No comments.

Additional comments

The authors mention that the cuticle of cockroaches would need resistance to puncture. However, what are the main predators of the cockroaches in their natural habitate? Are they mainly hunted/killed by puncturing predators? I guess that the thickness and puncture resistance of cuticle would for example not be a major factor against rodents or birds. Please comment on this.

Could the authors please provide a citation for the statement on the role of the exo- and endocuticle ( line 391ff) "The exocuticle consists of the sclerotized portion of the cuticle that grants the cuticle its rigid characteristics while the endocuticle consists of softer connective tissues that provide the cuticle its ability to restrict or permit deformation under tensile stresses and strains."

The sentence starting line 406 "By that measure, P. americana runs four times faster than a cheetah (Heinrich, 2001), which might not be possible with thicker, more protective cuticle." As the authors stated before, the scaling of bodymass is a very important factor, in particular when comparing a cockroach with a cheetah. Please discuss or rephrase. Also, I don't think that the added mass of a thicker, more protective cuticle would notably slow down a cockroach. If there is no data available from the Full-lab, maybe the authors could do a quick tests and just add some additional mass to the cockroach and record the fastest running speed?

Also see line 415 "the idea that thicker (and, in this case, stronger) cuticle may impede running speed." Please rephrase this part of the discussion or provide experimental data/citation.

Reviewer 2 ·

Basic reporting

No Comments

Experimental design

No Comments

Validity of the findings

No Comments

Additional comments

This manuscript reports the mechanical properties (puncture resistance, tensile strength, peak strain, and strain energy storage) of thoracic and abdominal cuticle from three cockroach species to see if there is a correlation between cuticle strength and predator avoidance behavior. The hypothesis is that cockroaches with less developed response will have stronger cuticle as a protective measure, then those with a greater avoidance behavior in which a less stiffer cuticle may be more conductive to movement and fleeing. The experiments were competently performed with adequate replicates and statistical analysis. The manuscript is well written, especially as it is targeted for a general audience that may not be familiar with the different mechanical concepts. I have one major point that I believe needs to be addressed (was the cuticle tested fully developed?) and a few minor points that could benefit from clarification. I recommend publication after these revisions.

1) No mention is made of the age of the adult insects used for the experiments. I would like some information on how long it takes after adult eclosion for the cuticle to fully mature. Cuticle synthesis and sclerotizaion may continue for several days or weeks before it is complete. If testing is done before this process has finished then the mechanical properties or cuticle thickness reported may not accurately reflect true values. I think being able to address this question, that testing was performed on fully developed cuticle, is important if the goal is to correlate cuticle properties with behavioral escape responses. (What if P. americana cuticle from slightly older adults have much stronger mechanical properties or cuticle thickness? This could have a significant impact on supporting or rejecting the hypothesis.)

2) I am perplexed by the notion that tensile strength and strain energy storage may be correlated more so to cuticle thickness than composition (lines 376-378). Since force values are normalized to cross sectional area in order to calculate stress, isn’t variation in cuticle thickness accounted for when determining tensile strength and strain energy? Therefore, wouldn’t increases in these values be more likely due to cuticle composition?

3) I would like the authors to consider using a different term than “composition” when referring to the different morphological layers of the cuticle (morphology? architecture?). To me, “composition” suggests biological components such as protein content and chitin; differences which were not determined in this study.

4) I also do not like the use of the term “connective tissues” (line 392) to describe the endocuticle. The main difference between exocuticle and endocuticle is likely the degree of sclerotization that occurs, being more so in exocuticle and less or absent in edocuticle. Connective tissue in insects is composed of collagen fibers, elastin, glycosaminoglycans, and glycoproteins. I’m not aware of either collagen or elastin having been detected in insect cuticle, although I may be wrong. While glycoproteins have certainly been detected in insect cuticle, I don’t know if they have been shown to be restricted to the endocuticle. Muscle fibers have also been shown to be present in cuticle but this is not surprising as one of the functions of the cuticle is as an attachment site for muscle, however, I do not know if it is known that they are restricted to the endocuticle. I would suggest that the authors drop the term “connective tissues” and instead use “unsclerotized layers” to describe the endocuticle.

5) The authors refer to P. americana as having “lighter-weight” cuticle (line 402) but they actually never determined the weight of the cuticle. Since G. portentosa is larger it will have more cuticle and thus more mass but it also has more muscle in which to lift that weight. If the weight per volume of cuticle was not determined then I question the use of “lighter-weight” as a descriptive term.

---

## Round 0.2 · accepted · Accept

· Academic Editor

Accept

In my determination, and affirmed by one of the reviewers, you have adequately addressed previous minor concerns with the manuscript.

Reviewer 1 ·

Basic reporting

No Comments

Experimental design

No Comments

Validity of the findings

No Comments

Additional comments

All of the comments have been sufficiently addressed.